# Structural Health Monitoring of Adhesively Bonded Pipe-to-Socket Joints by Integration of Polymer Optical Fibers and Their Load-Dependent Transmission Properties

**DOI:** 10.3390/s23104748

**Published:** 2023-05-14

**Authors:** Josef Weiland, Billy Kunze, Michael Luber, Naomi Krüger, Alexander Schiebahn, Rainer Engelbrecht, Uwe Reisgen

**Affiliations:** 1Welding and Joining Institute, RWTH Aachen University, Pontstraße 49, D-52062 Aachen, Germany; 2Polymer Optical Fiber Application Center, Technische Hochschule Nürnberg Georg Simon Ohm, Wassertorstraße 10, D-90489 Nürnberg, Germany

**Keywords:** adhesive, pipe-to-socket joint, SHM, polymer optical fiber, optical sensors

## Abstract

Adhesively bonded pipe-to-socket joints are used in numerous industrial applications. One example is in the transport of media, e.g., in the gas industry or in structural joints for such sectors as construction, wind energy, and the vehicle industry. To monitor such load-transmitting bonded joints, this study investigates a method based on the integration of polymer optical fibers into the adhesive layer. Previous methods for monitoring the condition of pipes, such as acoustic or ultrasonic methods or the use of glass fiber optic-based sensors (FBG or OTDR), are very complex in methodology and require cost-intensive (opto-) electronic devices to generate and evaluate the sensor signals; they are therefore unsuitable for large-scale use. The method investigated in this paper is based on the measurement of integral optical transmission with a simple photodiode under increasing mechanical stress. When tried at coupon level (single-lap joint), the light coupling was varied to obtain a significant load-dependent sensor signal. Based on an angle-selective coupling of 30° to the fiber axis, a drop of 4% of the optically transmitted light power by a load of 8 N/mm^2^ can be detected for the adhesively bonded pipe-to-socket joint with the structural adhesive Scotch Weld DP810 (2C acrylate).

## 1. Introduction

Adhesive bonding technology is becoming increasingly important in industry today. This is due to the fact that adhesive bonding has a number of advantages over other joining methods, such as welding [1]. More specifically, this technology results in good stress distribution on the pipe-to-socket joint and the replacement of materials from metals to polymers (fiber reinforced polymers (FRP)) because of their manufacturing benefits, lower weight, and good corrosion resistance [2]. Adhesively bonded pipe-to-socket joints are therefore used in numerous industrial applications including the transport of media [3] (e.g., pipelines on offshore oil platforms) [2] and in structural joints (e.g., in construction [4,5], wind energy [6], and the vehicle industry) [7]. There is a growing need for monitoring methods as the use of adhesive bonding technology in pipe construction increases. This is especially true when significant personal injury, environmental pollution, or financial loss could result from pipe failure. Various methods for monitoring bonded joints have been described in the literature. A variety of these methods are based on nondestructive testing techniques and have been further developed for structural health monitoring (SHM). For adhesive bonded joints, these methods can be classified into impedance spectroscopy [8], ultrasonic methods [9], back-face strain [10,11], and adhesive layer strain [12].

In the current literature, there are numerous publications on the integration of optical fibers into adhesively bonded joints to measure the strain of the adhesive. Typically, fibers made out of silica glass (amorphous SiO_2_) are used, in contrast to our study which used polymer optical fibers (POF) [12,13,14,15]. Most publications on silica fiber optic sensors describe either single-point sensors such as fiber Bragg gratings (FBGs) [16,17], quasi-distributed sensors (multiple FBGs), or full fiber-length distributed sensors such as optical time domain reflectometers (OTDR) and coherent optical frequency domain reflectometry (c-OFDR) [18,19].

Among these methods, fiber Bragg gratings (FBG) in silica fibers are routinely used for structural health monitoring in general. FBG are periodic structures in the light-guiding core of the fiber with a typical length of a few mm, reflecting light at a particular Bragg wavelength. This wavelength shifts deterministically by applying axial strain to the short FBG region of the fiber, providing a point-type strain sensor by measuring the change in the reflected wavelength with an optical spectrometer. Very small amounts of strain can be measured with FBG, typically in the order of a few microstrain. One microstrain written as 1 με is a relative change in length of $10^{-6}$. The Bragg wavelength of an FBG is sensitive to temperature, too, allowing temperature measurements, on the one hand, but introducing cross-sensitivity from temperature to strain measurements, on the other hand.

Grundmann [16] investigates the influence and suitability of FBG for the structural health monitoring of adhesive bonding with film adhesives in aviation. Stiffness-related measurement data are obtained from single-lap joints made of CFRP and an integration of the fiber in the direction of the force. In combination with other sensors, e.g., piezoelectric sensors, monitoring the integrity of the bond is possible. This is shown by the example of a shafted repair bond of a CFRP component. In [17], Fevery investigated the possibility of monitoring the expansion of adhesives by water absorption using embedded FBG. In this study, single-lap joints of aluminum with an epoxy adhesive are used. To correlate the results, an additional fiber outside the bond to compensate for temperature and material expansion effects is used. With this method, it is possible to make the expansion processes visible. At the end, the method is shown on a roof tile with an integrated solar module. In 2017, Shin et al. [20] investigated the orientation angle and sensitivity of FBG in fatigue testing. For this purpose, simple overlap bonding of aluminum with epoxy resin adhesive was used. The result is that the sensor sensitivity is highest when the fiber is embedded longitudinally to the load axis (0°). With 45° and especially 90° embedded fibers, detection of fatigue damage is difficult or impossible. In 2021, Shin et al. [21] investigated the full spectral response of the FBG sensor in fatigue testing with the same specimen geometry. He used a composite material and the same adhesive. For this purpose, he integrated the fiber in the direction of the force (0°). It was shown that the load-induced damage within the composite is manifested by peak shift, peak splitting, appearance of side peaks, and broadening of the FBG spectra. Using the same geometry, Shin et al. [22] investigated the hygrothermal behavior of FBG in 2022. For this purpose, the specimens were hygrothermally aged at 60 °C for 30 days and then subjected to cyclic fatigue testing. The premature damage could be detected in the spectrum of the FBG. The study by Schulz et al. [23] is one of the few that deals with shear strain acquisition. They investigated the embedding of multiaxial FBG in double-lap joints. Focusing on aerospace applications, aluminum joints are bonded with the epoxy resin. The fiber orientation is always 90° to the force application. Tilted FBG are used where the refractive index modulation planes of the FBG are tilted in respect to the fiber axis. Additionally, these planes are rotated in respect to the fiber axis and the load direction by 45°, 90°, and 135°. It is concluded that such multiaxial tilted FBG strain sensors embedded in an adhesive bonded joint can measure shear strain when the orientation of the FBG is aligned with the shear direction. However, the fabrication of such tilted FBG and the precise integration of the fibers are very complex. Cabral [2] investigated the use of FBG sensors on bonded FRP pipes on offshore oil platforms.

As mature as FBG sensing is for many applications, the need for a spectral measurement of the Bragg wavelength results in costs of the sensor interrogators in the order of €1000, up to a few €10,000 for interrogators that can address many 100 FBG in different fibers. Silica FBG are much stiffer than typical adhesives, which makes it difficult to quantify which amount of the load of the adhesive joint is transferred into and measured as strain at an embedded silica fiber. FBG are the point sensors most sensitive to axial strain, which requires embedding the FBG parallel to the load direction and perpendicular to a line of an adhesive joint. Monitoring a long length of an adhesive bond would require meander-like geometry of the sensing fiber, placing an FBG at every crossing point with the adhesive. Finally, for typical loads and deformations of adhesive bonds, microstrain sensitivity is not required.

It is the goal of our study to demonstrate a low-cost fiber optical sensor using polymer optical fibers (POF), which can be interrogated by low-cost optical light sources and optical power measurements, potentially enabling a broad field use. As described in Section 2, the fiber can be embedded along the line of the adhesive bond and provide an integral detection of load conditions of the adhesive joint. This is enabled by the mechanical properties of a POF, which are very similar to some typical adhesives, in contrast to very stiff silica fibers. The basic principle and applications of POF sensing to coupon test specimens have already been examined in our previous study [12,13,14,15].

Apart from our own study, no other published work has been conducted on the integration of polymer optical fibers into adhesives for structural health monitoring. However, the potential applications of the low-cost, intensity-based POF for SHM in polymeric materials could be confirmed theoretically and experimentally on glass fiber–reinforced polymers, where the sensor fibers are loaded in the axial direction. For example, the research group of Seyam and Peters [24,25,26,27] investigated the integration of POF into woven GFRP structures and its use as an intensity sensor. The POF was centrally integrated into the GFRP, but the layer structure varied. The samples were subjected to a 3-point bending test, and the change in transmittance as a function of force was measured. At the beginning of the test, the transmittance change was linear for all the tests. As the force increased, the transmittance decreased exponentially. When the structure failed, the transmittance dropped significantly toward 0%. This indicated destructive damage to the POF. The optical behavior of the POF was comparable for all the samples but shifted to higher forces as the number of layers increased. However, up to now, there is no published study that has addressed the monitoring of the load and structural health of a bonded pipe-to-socket joint using POF.

## 2. Materials and Methods

### 2.1. Fundamentals

A polymer optical fiber (POF) is made of a transparent core surrounded by a transparent optical cladding. Light is guided inthe core of the POF by the principle of total internal reflection if the necessary condition $n_{co}$ > $n_{cl}$ is fulfilled for the refractive indices of the core $n_{co}$ and the cladding $n_{cl}$. The refractive index is a material property defined by the ratio of the speed of light propagating in this material in relation to the speed of light in vacuum. Light beams from a light source passing the surrounding air and entering the fiber core at an angle Θ in respect to the fiber axis are refracted at the air-core interface according to Snell’s law. The light is guided inside the core if these refracted light beams meet the condition for total internal reflection at the core–cladding interface. In the approximation of ray optics—which holds here, as the core diameter is much larger than the wavelength of light—a particular propagation angle of the light beam in respect to the fiber axis can be identified with a particular geometry of the electromagnetic field of a propagating light wave, called a mode. Many individual modes (angles) of light can propagate within the fiber core due to the large fiber diameter relative to the wavelength. To fulfill the condition of total internal reflection, the angle of the rays entering the fiber from the air must not exceed the acceptance angle $\Theta_{max}$. This angle is usually expressed by its sine, which is called numerical aperture NA, given by the refractive indices of the core and the cladding by [28]:(1)$$NA=\sin\Theta_{max\;}=\sqrt{n_{co}^{2}-n_{cl}^{2}}$$

For this study, a POF with a numerical aperture of NA=0.5 has been used, which corresponds to an acceptance angle $\Theta_{max}=30{}^{\circ}$ for light coupled into the fiber.

The adhesive and the POF (both of which are polymers) have similar mechanical properties, and this is what enables the method of using polymer optical fibers for SHM. When an external force is applied, the integrated POF is subjected to the strain of the adhesive layer. By integrating the POF at 90° to the direction of the force, the POF is stressed radially. This causes geometrical changes, strains, and stresses in the optical material and the core–cladding interface. This stress changes the optical transmission properties of the fiber. Some of the light rays will violate the total internal reflection conditions, and the light will be emitted from the cladding material. Figure 1 shows this schematically. These changes in light propagation can be detected by optoelectronic sensors at the end of the fiber. As a function of the bond load, the resulting sensor signal is evaluated. In this way, the load state of the bond can be monitored in a nondestructive manner. Compared to FBG sensors, the POF method represents an integral sensor over the entire embedding length. This is possible with simple optoelectronic measuring devices but does not provide local resolution. Interestingly, as this sensor principle relies on a significant transversal deformation of the fiber, it is not sensitive to temperature changes, in first place. However, temperature may affect the mechanical properties of both the POF and the adhesive.

### 2.2. Polymer Optical Fiber

A polymer optical fiber (POF) type “DB-500” from Asahi Kasei (Tokyo, Japan) with a nominal diameter of 500 ± 23 μm is used for the experiments. The POF used has a light- conducting core of polymethyl methacrylate (PMMA) (diameter 480 μm). This core is surrounded by a 10 μm thick fiber cladding. The cladding is made of a blend of polyvinylidene fluoride (PVDF). Figure 2 shows experimental data from the tensile tests of the individual fiber components, the entire POF, the core (PMMA), and the cladding (PVDF). According to Equation (2), the Young’s modulus of the fiber is 2668 N/mm^2^. The Young’s modulus of the core material PMMA is calculated to be 2996 N/mm^2^, while the Young’s modulus of the cladding material PVDF is 847 N/mm^2^. Figure 2 shows that the PVDF cladding has a significantly lower stiffness than the PMMA fiber core. This can also be seen in the publication by Kuang et al. [29].

### 2.3. Adhesive

The two-part acrylic adhesive 3M Scotch-Weld DP810 is used in this study. This adhesive is capable of bonding both polymeric materials and metal components. Çitil et al. [3] used this adhesive to repair steel pipes in a high-pressure application. The result of the tensile test according to DIN EN ISO 527 is shown in Figure 2. The Young’s modulus of 1094 N/mm^2^ is then determined analytically using Equation (2), where *σ*_1_ and *σ*_2_ are the stresses measured at *ϵ*_1_ = 0.0005 and *ϵ*_2_ = 0.0025
(2)$$E=\frac{d\sigma }{d\epsilon }=\frac{\sigma_{2}-\sigma_{1}}{\epsilon_{2}-\epsilon_{1}}$$

### 2.4. Manufacturing Process

The adherent material of the single-lap joint specimen is a cold-rolled deep-drawing steel DC04 from Thyssenkrupp Steel Europe AG (Duisburg, Germany). It is coated with a cathodic dip coating Cathoguard 800 from BASF Coating GmbH (Münster, Germany). The steel specimens are 3 mm thick, 100 mm long, and 25.4 mm wide. The overlap length of the bond is 12.5 mm, and the adhesive layer thickness is 3 mm. The POF is embedded in the center of the joint with an embedment length of 25.4 mm. The parts to be joined are pre-treated in an atmospheric pressure plasma process using the Tigres Plasma Blaster MEF from Tigres GmbH (Marschacht, Germany). To pretreat the POF or the cladding, a low-pressure plasma process is used with the Tetra-5-LF system from Diener Electronic GmbH + Co. KG (Ebhausen, Germany). The curing time is at least 7 days.

In Figure 1 on the right side is shown a schematic view of the pipe-to-socket joint with an integrated POF. The pipe is made from PMMA and has an outer diameter of 56 mm; the socket is made of PLA, of which the inner diameter is 62 mm. The overlap length is 10 mm long, and the thickness of the adhesive layer is 3 mm. The dimension of the adhesive layer during the manufacturing process is controlled by a space holder. The POF is wired spiraled around the inner pipe and should be in the middle of the adhesive layer. To secure the position of the POF, the inner pipe is 3D printed and has positioning aids for the POF. The embedded length of the POF is about 186 mm. Before joining, each part of the pipe-to-socket joint is cleaned and pretreated with the same vacuum plasma process mentioned above. The adhesive is injected between the inner pipe and the socket. The bond is cured for 7 days at room temperature.

To ensure lossless and industrially usable light coupling and decoupling of the POF, the ends are fitted with field-installable subminiature assembly (FSMA) connectors and the end surfaces are ground (grit P1000) and polished (grit P4000) in two steps. In our laboratory work, the fiber was a bare fiber with an optical core and cladding, both of polymer, with no additional protective coating. As polymers, these bare fibers are mechanically very robust, in contrast to bare silica glass fibers. For field use, protective coatings against UV solarization may be required for the fiber sections outside the adhesive from the light source to the detector.

### 2.5. Optical Test Setup

The test setup for the investigation of the single-lap joints can be seen in Figure 3 on the left side. For these experiments, the tensile testing machine Zwick RetroLine Z010 from Zwick-Roell GmbH & Co. KG (Ulm, Germany) is used. The identical specimen is cyclically loaded quasi-statically (1 mm/min) up to 3000 N. The method of light coupling in the POF is varied between the individual cycles, and the total optical power is measured when the light leaves the POF using a photodiode. An LED and a laser diode with 650 nm are used as the light source. By using the LED, a full excitation of the POF is generated, which means that the complete fiber end face is illuminated and all possible reflection angles are coupled in. With the laser diode, an angle-selective coupling takes place. To generate an angle-selective light coupling, the PL204 laser diode module from Thorlabs, Inc. produces a red (635 nm), collimated, circular beam with a diameter of 3 mm; a power of 0.9 mW is used. The laser module is mounted on a rotary stage with the fiber end face positioned at its center. The angle to the fiber axis can vary in the range of 0–40°. For the optical investigations, the angles 20°, 25°, and 30° are considered. Angle-selective excitation is used to guide light beams into the POF with a high angle of reflection (close to the critical angle of total internal reflection). These angles are more sensitive to optical changes in the core–cladding interface; moreover, they require many core–cladding reflections to get the distance from coupling to decoupling. The signal at the other end of the POF is measured by a PDA100A2 Si Switchable Gain Detector from Thorlabs GmbH (Bergkirchen, Germany) by converting the light power into electrical voltage. The detector records the values at a frequency of 15 Hz, which is sufficient for the quasi-static load case of pipe-to-socket bonding. The signal at the beginning without any loading is then set to 100% and shows the results as transmission in percent.

The test setup for the investigation on the pipe-to-socket joint can be seen in Figure 3 on the right side. For these experiments, the tensile testing machine Instron 4210 from Instron (Norwood, MA, USA) is used. The identical specimen is subjected to compressive force and cyclically loaded quasi-statically (1 mm/min) up to 15,000 N. For the pipe-to-socket joints, only the angle-selective light coupling with the laser is investigated. With only one light source and one photo diode required for pipe-to-socket bond monitoring, this is an extremely cost-effective solution.

## 3. Results

### 3.1. Single Lap Joint Cyclic Tests

The results of the investigations of the single-lap joints are shown in Figure 4. The figure shows both the cyclic force-displacement curve and the transmission-displacement curve resulting from the respective tensile test with different light coupling. It can be seen that the 20°, 25°, and LED coupling methods lead to only very small transmission changes of less than 1%. A more distinctive behavior can be observed in the graphs for the 30° coupling. A constant linear decrease in the transmission with increasing load is observed. The changes in transmission are between 1.5% and 3% when the specimens are loaded up to 3000 N.

### 3.2. Pipe-to-Socket Joint Cyclic Tests

The results of the investigation of the pipe-to-socket joint are shown in Figure 5. The force-displacement curve and the transmission-displacement curve for different couplings are shown. In the case of the pipe-to-socket joint, it can be seen that there is a very clear drop in transmission with increasing force, especially at the angles of 25° and 30°. For samples PSJ1 and PSJ2, the transmittance curves are almost linear. In the case of sample PSJ3, there is a steep drop in transmission as the displacement increases. This could be related to the larger displacement. For all the samples, the 30° coupling has the most significant optical changes, ranging from 2% to 5% change at a load of 15,000 N.

### 3.3. Comparison of Single-Lap Joint and Pipe-to-Socket Joint

Figure 6 shows comparative results between the single-lap joint and the pipe-to-socket joint. To compare the tests, the transmission at the resulting stress of the adhesive layer is shown. All the graphs are shown for the coupling at 30°. It can be clearly seen that the pipe-to-socket joints show a larger change at the same stress. The almost linear curve of both geometries has a larger slope for the pipe-to-socket joint. Considering the load of 5 N/mm^2^, the optical transmission change in the pipe-to-socket joint is 2–3 times higher than that in the single-lap joints.

### 3.4. Pipe-to-Socket Joint, Destructive Tests at Acceptance Angle

In addition to the cyclic tests on the single-lap joints and the pipe-to-socket joints, three other pipe-to-socket joints were also subjected to destructive testing. The most promising light coupling at maximal coupling angle (Θmax = 30°) was selected. The results are shown in Figure 7 as transmission versus stress curves. A linear drop in transmission can be seen even at low stresses. Up to a stress of ~10 N/mm^2^, the transmission curves decrease linearly to values between 97 and 94%. Thereafter, the decrease in transmission increases exponentially, and at ~13.5 N/mm^2^ the specimens fail. In the case of failure, the transmission of all the specimens is ~90%. In particular, the exponential drop in transmission is a clear sign of damage in the bond. This mechanical-optical behavior of the joint between the adhesive and the POF can be used for early detection of failure and thus for structural health monitoring of the bond.

## 4. Discussion

The results of the investigations show consistent optical and mechanical behavior. As soon as the fiber-integrated bond is subjected to an external force, the resulting stresses are transferred to the POF and change the optical properties of the POF. The evidence is provided by the change in optical transmission for all the tests. For all the tests, the coupling of light into the fiber near the maximal coupling angle (Θmax = 30°) resulted in the highest optical transmission changes. Thus, the assumption that especially the core–cladding interface is stressed and subsequently influences the optical transmission under multiple reflections can be strengthened. Furthermore, the comparison of the two geometries and the resulting different embedding lengths showed that the change in transmission increases with the length of the embedding. This can be seen in Figure 6, since a 2–3 times higher change in transmission was detected for the same resulting stress. The constant stiffness of the bonded joints can be observed in the force-displacement curves, which means that the mechanical impairment of the bonded joint by the POF can be considered low for this combination of adhesive and POF. The destruction tests show that up to ¾ of the maximum force there is a linear relationship between stress and transmission. From this point on, the transmission decreases exponentially up to 90%. From this behavior, an SHM method can be developed. This method takes into account the change in transmission over time. This can be carried out by determining the gradient in the transmission stress diagram or by determining the first derivative. In this way, a simple and robust SHM method is obtained. Overall, the investigations show that the method of structural health monitoring of bonded joints utilizing POF also has a high potential for complex geometries such as pipe-to-socket joints. The change in optical transmission can be detected with low-cost optoelectronic components, which is a major advantage compared to sensing with fiber Bragg gratings. This makes the method suitable for the integral structural health monitoring of the entire bond without spatial resolution in favor of a low-cost sensor for many industrial applications.

## 5. Conclusions

In the introduction, the publication discusses the growing number of pipe-to-socket joints in all industries. This is due to the change in the materials composing polymeric pipes but also to the advantages of adhesive bonding over other joining technologies. Since the failure of such joints can have significant economic and social consequences, there is a growing need for methods to monitor the integrity of bonded joints. Current methods, such as FBG, in combination with adhesives have the disadvantage of a high stiffness difference and, in particular, the fact that the FBG must be integrated in the force direction and cannot be positioned along the adhesive joint. The method based on the integration of polymer optical fibers offers great potential from both a mechanical and an economic point of view. The basic mechanisms have been demonstrated in other publications by the authors, but not in combination with an acrylic adhesive nor in the complexity of a pipe-to-socket joint. The experimental part of the paper shows that with an angular coupling of 30° into the POF and a simple measurement of the photocurrent with a photodiode, reproducible test results can be obtained. These are linearly related to the applied force. The transfer to the pipe-to-socket joint geometry was also successful. With the help of positioning aids, the POF can be integrated in the center of the bond and act as a sensor to directly measure the stresses. Due to the longer embedding length, a stronger correlation between stress and transmission can be seen. The destructive tests, with an exponential decrease in transmission at higher stresses, provide a simple but robust method for early failure detection when the derivative of transmission over time is taken into account. The integration of POF into the pipe-to-socket joint thus enables a method of structural health monitoring not previously considered by the state of the art and offers a high potential for use soon. In future studies, the change in transmission needs to be tested under other stresses, especially climatic stresses, which can have a negative effect on the optical properties. In these investigations, the aim is to compensate the change in the signal due to temperature by combining several optical effects, e.g., angle-dependent far-field analysis.

## Figures and Tables

**Figure 1 sensors-23-04748-f001:**
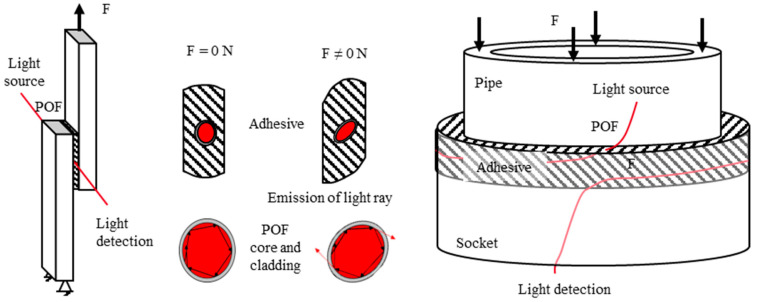
(**Left**/**right**): Schematic view of the single lap joint and the pipe-to-socket joint; middle: sensor principle—strain/stress transfer adhesive to POF.

**Figure 2 sensors-23-04748-f002:**
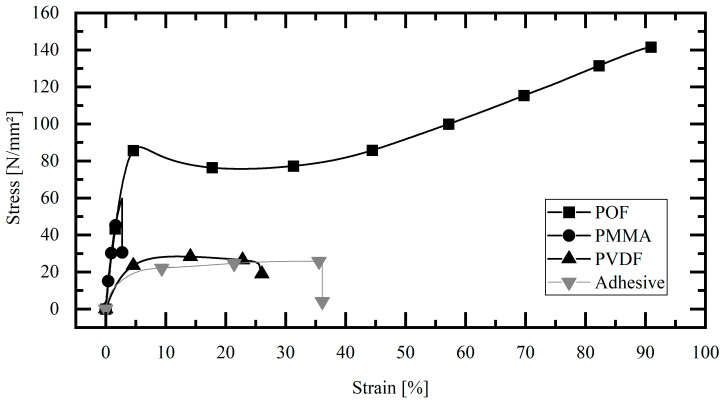
Tensile test adhesive and POF materials.

**Figure 3 sensors-23-04748-f003:**
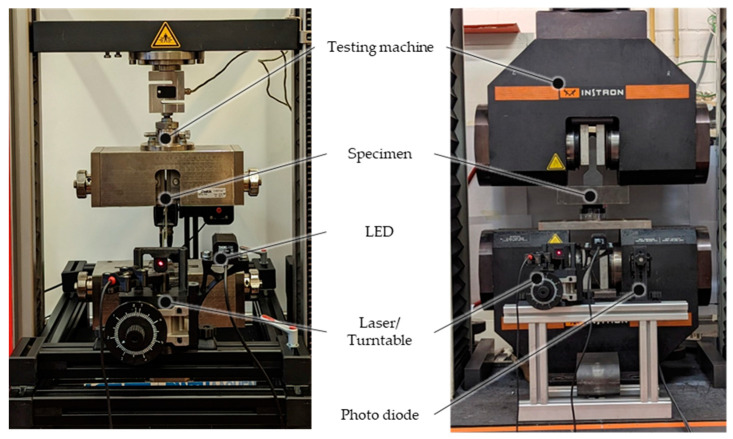
Optical test setup. (**Left**) single lap joint; (**Right**) pipe-to-socket joint.

**Figure 4 sensors-23-04748-f004:**
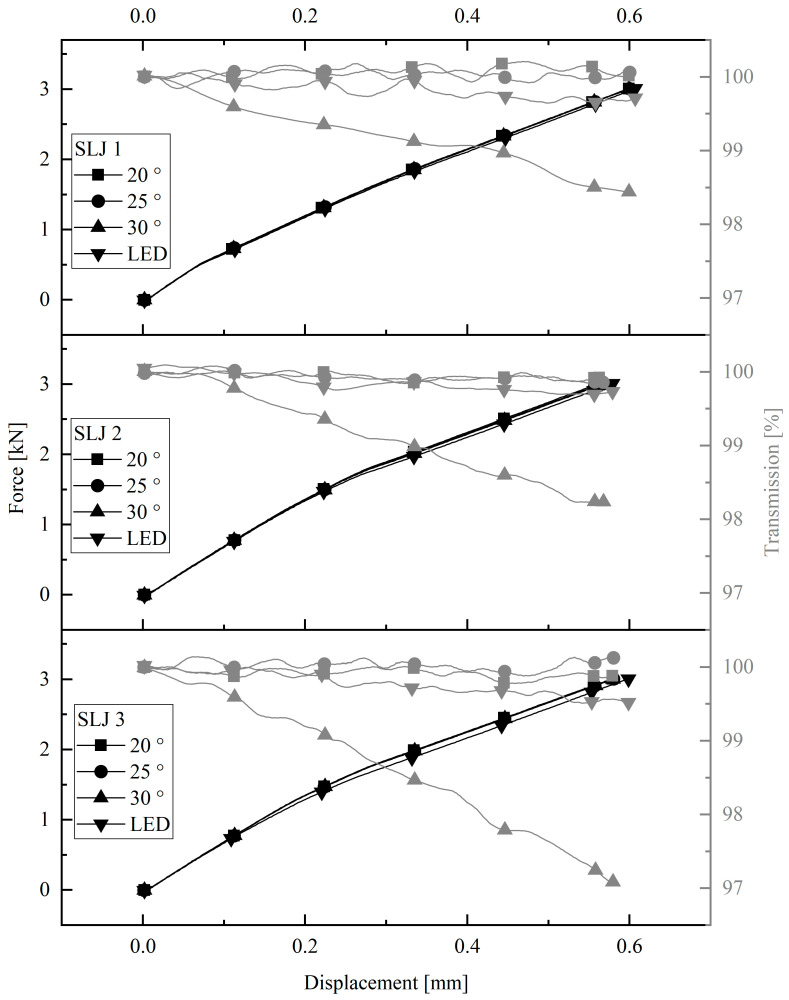
Optical results of single-lap joint.

**Figure 5 sensors-23-04748-f005:**
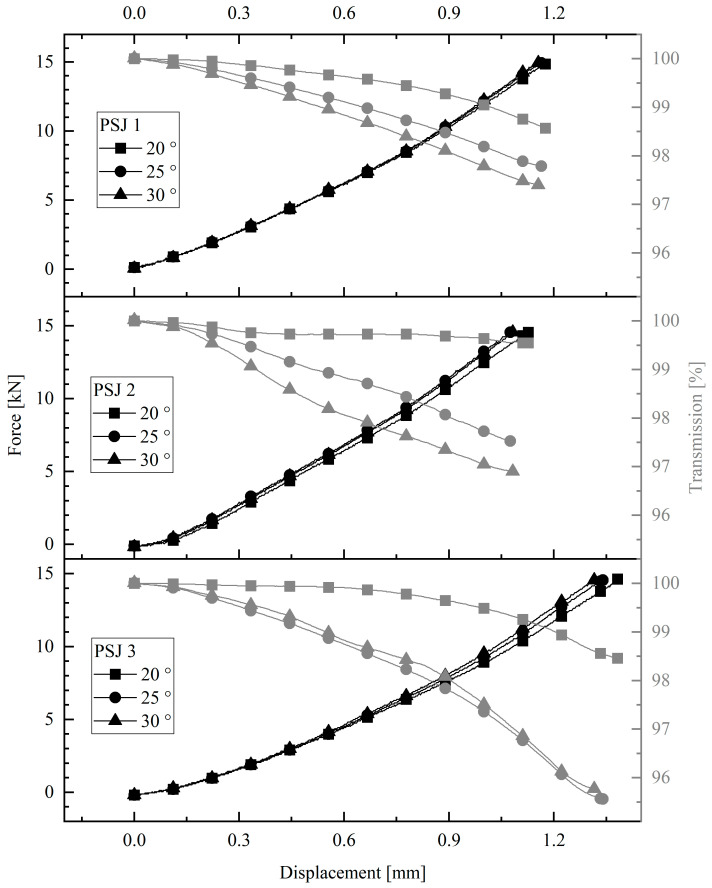
Optical results of pipe-to-socket joint.

**Figure 6 sensors-23-04748-f006:**
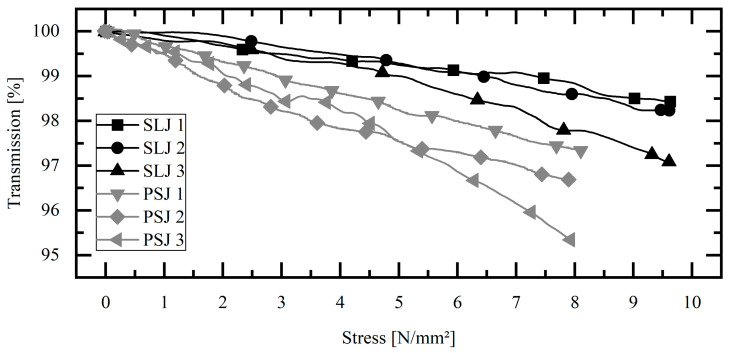
Comparison of optical results of single-lap joint and pipe-to-socket joint at Θmax = 30°.

**Figure 7 sensors-23-04748-f007:**
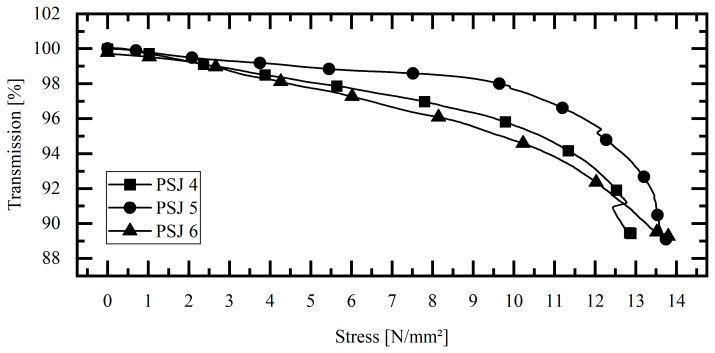
Optical results of pipe-to-socket joints—destructive tests at Θmax = 30°.

## Data Availability

Data are available from the authors upon request.

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
