# Peer review of "Structural Health Monitoring of Adhesively Bonded Pipe-to-Socket Joints by Integration of Polymer Optical Fibers and Their Load-Dependent Transmission Properties"

_sensors, 2023, doi:10.3390/s23104748_

Round 1
Reviewer 1 Report
This study proposes a polymer optical fiber(POF) based pipe-to-socket SHM method. This article provides a basic idea of a new sensing mechanism with experimentally validated mechanical properties for their setup.
1. The terms "n_co" and "n_cl" are not familiar to non-experts in optical sensors. Without more context, it's difficult to determine what these terms refer to. It would be helpful for the authors to provide a brief explanation or definition of these terms in the paper.
2. can you explain about sensing fundamental of POF more detail? What is main difference between FPG type sensors and POF? What is the typical sensing range for single CH. For example, typical FPG based strain sensing has 1~2mm of fiber bragg to sense single point strain and able to sense ~micro-strain.
3. For SHM purposes, it's important to characterize how the behavior of the adhesive and setup varies with temperature, as temperature changes can affect strain measurements. The authors should provide more information on how they addressed this issue, including any mitigation strategies used to minimize the impact of temperature variations.
4. The sensitivity of the setup in micro-strain level is not explicitly stated in the paper. The required sensitivity for this application will depend on the specific use case and desired accuracy. The authors should provide more information on the sensitivity of their setup and whether it meets the required sensitivity for their application.
5. The general requirements for sampling rate and resolution in pipe-to-socket joint SHM will depend on the specific application. The authors should provide more information on the sampling rate and resolution used in their setup and whether it meets the requirements for their application. The cost-effectiveness of the setup compared to off-the-shelf options should also be discussed. The number of channels required for monitoring a single pipe-to-socket joint and whether the setup is chainable like an FBG array sensing system should also be addressed.
6. The authors should consider adding clearer figures to the paper, especially for Figure 3, which depicts the pipe-to-socket joint test. A simplified figure with labels and annotations would be helpful for readers to better understand the test setup and results.
7. The authors should clarify their statement in the conclusion about FBGs not being commercially viable or requiring further research. While it's possible that FBGs may not be the best option for pipe-to-socket joint SHM, it's important to note that FBGs are widely used in various SHM applications and are commercially available. The authors should provide more context and justification for their claim.
This article is written clearly.
Author Response
The authors would like to thank reviewer #1 for the constructive comments.
Due to the fact that the method is a fundamentally new development of a sensor, the method cannot compete with the capabilities and sensitivities of an FBG at this point in time. However, it has the great economic advantage that the simplest opto-electronic devices are sufficient. From our point of view, this method can therefore be used in a wide range of applications. Nevertheless, further investigations are necessary, in particular the behavior of the sensor at different temperatures. Tests on this are being planned.
The individual notes of the reviewer have been revised to the best of our knowledge and are answered directly below.
- We have added a sentence to describe the refractive indices of n_co and n_cl. Line 137-139
-
Compared to FBG sensors, the POF method represents an integral sensor over the entire embedding length. This is possible with simple optoelectronic measuring devices, but does not provide local resolution. Line 165-166. In addition, section 1 has been rewritten and enhanced to make the differences of our POF sensors compared to silica fiber FBG more clear.
-
This is absolutely correct. however, as the method is a new method developed from the ground up, only the mechanical stresses have been investigated at this stage. In the next step, climatic stresses, especially temperature, will be examined intensively. This is described in the conclusion. Line 356-358. However, our sensor principle is not cross-sensitive to temperature in the first place, as now described at the end of Sec 2.1, in contrast to FBG.
-
The method chooses the approach that the whole POF serves as an integral sensor, this allows a cost effective method but with the omission of a local resolution. However, this is sufficient for the pipe to socket joint and the quasi-static load case shown. Additionally, the deformation of the joint reaches many percent before failure, so microstrain resolution as provided by FBG is not required, and out of the scope of our sensor.
-
Further information on the configuration has been added. Line 245-247. The setup requires only one POF for the pipe-to-socket bond each with light source and photodiode, which is an extremely cost-effective method.
- Figure 3 has been replaced by a new one. We hope that the experimental set-up is shown more clearly there.
- The section in the Conclusion has been rewritten to clarify the disadvantages of FBG in relation to adhesives
Reviewer 2 Report
This is an interesting paper demonstrating the potential of plastic multimode optical fibers to serve as load sensors with particular applications to adhesively bonded pipe-to-socket joints. The paper is well structured and professionally written. I have detected very few English issues, detailed at the end of this review.
The paper clearly demonstrates consistent correlation (within the scope of the reported experiments) between the load and the decrease in transmission of light through the embedded optical fiber. While below failure the transmission decreases linearly with the load,near failure it does so exponentially.
Here are a few issues, requiring the authors attention:
1) I am not 100% sure the word SHM is fully appropriate as the first word of the title. Yes, it appears to be a load monitoring technique (although more data should be collected on the effect of aging and temperature on the transmission even at a constant load). Can the technique detect deterioration of the bond, not only when the load is close to failure? In fact, the authors themselves discuss the potential of the method to become an SHM method (see around line 275 and below).
2) More details should be provided about the ingress and egress points if the fiber is required to be functional for a long time in an industrial environment.
3) The angle for good mode coupling may depend on the diameter of the winding and on the NA of the fiber.
4) For the method to succeed it is very important that the fiber jacket quickly absorbs cladding modes
5) Comments:
Line 17: OTDR instead of ODTR
Line 48: "...the optical light guiding properties in dependence of the external". I prefer: the dependence of light guiding on the external load
Line 165 Therefore (instead of Therefor)
Line 187: "The light coupling into the POF is varied for each cycle" -What do you mean here?
Line 207: "The identical specimen is subjected to pressure, cyclically loaded quasi-statically (1 mm/min) up to 15000 N". What pressure? I thought you apply tension. And if 'pressure' why the unit are those of force?
Line 256: "This mechanical-optical behavior of the joint between the adhesive and the POF can be used for early detection of damage or failure of the bond." You did NOT show DAMAGE detection, only under failure.
Line 268: "Furthermore, it could be proven that the change in transmission increases with the embedding length". How proven? Reference?
Line 277: "This can be through the determination of the". English
I mentioned them in my comments to authors
Author Response
The authors would like to thank reviewer #2 for the constructive comments.
We would like to thank you in particular for the clarification and classification of the method with regard to failure and damage. From our point of view, the results shown indicate that a failure is detected at an early stage in the loading scenario shown. Since the method is being developed from scratch, further investigations are of course necessary, especially with regard to the detection of local damage (e.g. cracks), and the effects of temperature or climate in general will have to be considered in the next stages.
The individual notes of the reviewer have been revised to the best of our knowledge and are answered directly below.
- From our point of view, the monitoring of the force and the resulting failure can be assigned to Structural Health Monitoring.
-
In our laboratory work, the fiber has been a bare fiber with an optical core and cladding, both out of polymer, with no additional protecting coating. As polymers, these bare fibers are mechanically very robust. For an field-use, protective coatings against UV solarisation may be required for the fiber sections outside of the adhesive from the light source to the detector. This has been added at the end of sec. 2.3
-
The maximum acceptance angle is significantly dependent on the NA. Thus we used light coupled into the fiber at an angle of 30° close to the NA. This modal distribution is remarkably stable over a few meters of a POF, unless the winding diameter gets too small. However, the winding will enhance coupling to higher order modes, which are the desired modes four, our sensing principle.
-
In this method, POFs are used without an outer jacket. Only the optically necessary cladding is still on the fiber - which is then in direct contact with the adhesive, and this absorbs the light rays.
- Comments
Line 17: check
Line 48: check
Line 165: check
Line 187: The sentence is rewritten
Line 207: In this context, pressure stands for a compressive force - the sentence is rewritten
Line 256: You are correct - damage was removed from the sentence
Line 268: Furthermore, the comparison of the two geometries and the resulting different embedding lengths showed that the change in transmission increases with the length of the embedding.
Line 277: The sentence is rewritten
Reviewer 3 Report
The article presented by the authors, in my opinion, is quite interesting and has some practical value. The methods are well described, the text is nicely designed, the conclusions are adequate. However, I have serious doubts about the advisability of publishing this manuscript in Sensors. Firstly, light-transmitting sensors are a well-known technology from a scientific and engineering point of view. Secondly, the use of polymer optical fibers as sensors are also well known. The conclusions of the authors state that on the basis of the data obtained by them, it is possible to develop a method. If the method had already been developed and described in this article, the probability of publication in Sensors would be higher. At the same time, it is obvious that the authors have done a great job and it should not remain unpublished. Therefore, I suggest that the authors resubmit the article to a journal of a lower rank, and after the method is fully developed and described in detail, resubmit the work to Sensors.
Perhaps my comments below can make this manuscript a little better:
1. In the abstract, the letters in the abbreviation OTDR are mixed up.
2. Almost all chapters of the article end with drawings. I suggest placing the drawings immediately after the first mention, and placing their descriptions below.
3. The fitting spline in figure 7 is chosen very strangely. Please note the behavior of the curve.
4. It is desirable to provide the parameters of optical fibers and summarize them in a table.
As for me it is OK.
Author Response
The authors would like to thank reviewer #3 for the constructive comments.
Reviewer #3 is correct in saying that there are optical fiber-based sensors that are widely studied in principle and already in industrial use. This refers in particular to FBG and OTDR sensors. However, the combination of these sensors for bonded joints or even integrating them into bonding has been very little considered so far. When this is done, it quickly becomes clear that the systems have significant disadvantages to be integrated into bondings. One of the biggest disadvantages is the mechanical behavior of the fiber. For example, glass has a Young's modulus of 80-90000 MPa and the adhesive has a maximum of 3000 MPa. This leads to extreme stress peaks. One novelty of our work is a dedicated modal excitation close to the NA of the fiber as described in sec. 3, which makes our sensor based on multimode polymer optical fiber sensitive to transversal deformation, in contrast to usual silica-glass based fiber sensors like fiber Bragg gratings, sensitive to axial forces. This has been addressed in a revised introduction in section 1. Due to these circumstances, this publication deals with a fundamentally new development of a fiber optic sensor and its first applications for structural health monitoring. The authors are aware that further investigations are necessary, especially on the method but also on the stresses, e.g. climate. These are planned for the future. Nevertheless, this publication with the shown content represents a considerable added value, on the one hand for the community of fiber optic sensors and on the other hand for the application in bonded structures and therefore justifies from our point of view the publication in the very respected Sensors journal.
The individual notes of the reviewer have been revised to the best of our knowledge and are answered directly below.
- The word is rewritten
- At the moment, the figures are only placed so that they waste little space. The position of the figures can be changed during the editing process.
- The changes in the sensor signal mean that this is due to the compressive load of the pipe-to-socket joint when it settles, because it is not possible to produce two parallel surfaces.
- The section on POF has been rewritten and should now be easier to read.
Round 2
Reviewer 1 Report
The authors well addressed the reviewer's concern.
Reviewer 3 Report
I thank the authors for their careful revisions, now the paper looks much better.
Now they have clearly highlighted their contribution and the novelty of this study, so they have convinced me that it is now suitable for the publication.